# The Influence of Laser Modification on a Composite Substrate and the Resistance of Thin Layers Created Using the PVD Process

**DOI:** 10.3390/s20071920

**Published:** 2020-03-30

**Authors:** Ewa Korzeniewska, Mariusz Tomczyk, Maria Walczak

**Affiliations:** Institute of Electrical Engineering Systems, Faculty of Electrical Engineering, Electronics, Computer and Control Engineering, Lodz University of Technology, 90-924 Łódź, Poland; mariusz.tomczyk@p.lodz.pl (M.T.); maria.walczak@p.lodz.pl (M.W.)

**Keywords:** textronics, physical vapor deposition, PVD, laser modification, design of experiments, wearable electronics

## Abstract

For physical vapor deposition (PVD) technology, cleaning a substrate is one of the key preliminary processes before depositing the metal layer. In this article, we present the results of research on the modification of a textile composite substrate using laser technology and its influence on the surface resistance of silver structures intended for use in wearable electronics. As a result of the substrate modification, the resistance of the layers increased as compared with the structures produced on an unmodified substrate. An experimental planning technique was used to optimize the laser modification process.

## 1. Introduction

The range of textile applications is constantly growing. This increase is due to the development of new raw materials, the invention of new technologies related to textiles, and the methods of their modification. The challenges posed to modern clothing are to provide thermal comfort and to also use the clothing as a decorative element. The uses for textiles depend on their functionalization. Surface coatings have provided an innovative contribution to the development of textiles and are particularly important with respect to the application. Textronics is one area of science that is developing based on the functionalization of textiles by combining textiles with computer science and wearable electronics. To create electrical circuit elements on textile products, various techniques have been used, such as physical vapor deposition (PVD), chemical vapor deposition (CVD), digital printing method, plasma-assisted physical vapor deposition, polymer-assisted physical vapor deposition (PAPVD), embroidery, etc. [1,2,3,4,5,6]. Another method for producing thin conductive layers on various substrates is the electrospinning technique [7,8]. As it turns out, this technique is suitable for the production of polymer nanofibers and also for the production of conductive layers that are widely used in electronic components and devices [9,10]. Electrospinning is also used to produce electrostatic, conductive nanofibers and transparent conductive layers [11,12].

In this study, we focus on the production of thin electroconductive layers using physical vapor deposition as an environmentally friendly process that is important for the improvement of corrosion and wear resistance [13].

Using PVD technologies, it is possible to produce coatings with a homogeneous structure, and layers with graded properties and controlled morphology, as well as coatings with excellent adhesion or with predefined shapes [14,15,16,17,18]. The process of physical deposition of layers has been improved by the results of many works on mathematical modeling and numerical simulations [13,19,20]. As a result, it is expected that the costs of carrying out the process could be reduced, as well as the impact on the quality and the mechanical and electrical properties of the structures produced in this process [21,22,23]. PVD coatings are also considered, for example, as an alternative replacement for hard chromium plating in the plastic mold industry [24].

The deposition of metallic layers on the surface of textiles requires initial preparation of the substrate in order to improve the adhesion of these layers. The Taguchi experimental design technique was used for choosing the best parameters of the laser beam (wavelength 1060 nm of radiation), for the required resistivity of the deposited layer. We present our results of investigating the impact of modifying the surface of a textile composite substrate using laser processing on the surface resistance of electroconductive structures obtained in the process of physical vapor deposition.

## 2. Materials and Methods

### 2.1. Physical Vapor Deposition

Evaporation is one of the techniques for surface coating in the PVD process. This technique releases metal particles which have been placed in a heated source and settles them on a substrate located in close proximity [13]. The substrate temperature is much lower than the temperature of boiling metal. A thin layer is formed as a result of atom-on-atom deposition on the prepared substrate. Individual particles of deposited metal are released by thermal evaporation from a resistance source. The thin layers usually have thicknesses of several microns. In this technique, the surface properties change and the transition zone between the substrate and the deposited material is observed. The vacuum environment inside the process chamber during deposition is conducive to the elimination of gaseous pollutants, and thus greater chemical homogeneity of the deposited layer. The substrate on which the thin electroconductive layer is deposited also affects the electrical and mechanical properties of the deposited layer [13]. A diagram of this PVD process technique is presented in Figure 1.

During the PVD process for creating thin films, toxic chemical precursors are not used, and no toxic reaction gas or liquid residual products are produced during the deposition process. Therefore, this PVD process is considered to be an environmentally friendly deposition process as compared with the chemical vapor deposition (CVD) process and even a plasma-enhanced chemical vapor deposition (PECVD) process. However, one should remember that due to the way the thin film is formed during the PVD process, and in particular due to the line-of-sight transfer of the vapor flux, the metal coatings, deposited with this technique, have many intrinsic defects (including columnar structures), cracks, discontinuities, as well as pinholes and pores, which can significantly affect their resistance. The quality of the layer can be verified using the non-invasive methods as [1,25,26,27]. The created layers should also be resistant to electromagnetic disturbances [28].

In the case of the conducted research, thin electroconductive metal layers were created using the PVD process on a textile composite substrate known in the market under the trade name Cordura (Miranda Ltd., Turek, Poland). Cordura is made of polyamide fibers which are coated with a polyurethane layer. A cross-section of the material is presented in Figure 2. The surface weight of the material is 380 g/m^2^. In the clothing and sports industry, the studied textile product is used as a material with very high mechanical resistance [4]. The creation of a thin metal layer on the surface gives this textile product a new functionality, i.e., it can be a substrate for the wearable electronics systems. For this research, during the process of creating the thin metallic structures, the following parameters for the PVD process were used:Initial vacuum, 5 × 10^−5^ mbar;Time of metal deposition, 5 min;Deposited metal, Ag with the purity of 99.99%.

The geometry of the created samples is presented in Figure 3. The internal electrodes are the voltage electrodes and the external are the current ones. The tested area was 50 mm in length and 3 mm wide.

### 2.2. The Laser Modification of Textile Composite Substrate

Laser surface modification using a highly energetic laser beam in a complex way is currently the most modern method of material engineering, providing not only extensive possibilities for changing the surface layer properties but also great precision during the process. Methods of ablative removal of layers with laser radiation allow precise shaping of conductive elements, or microfluidic structures, sensors, etc. [29,30,31,32]. Laser modification is widely used in many industries. Such modifications are used in the engineering industry, inter alia, for hardening various elements, texturing engine parts to improve lubricating, or for adhesive properties; in the textile industry, to improve utility and decorative fabrics; in the semiconductor industry, to increase the efficiency of solar cells [33,34,35,36]; and in electrical engineering, to obtain the required value of electric parameters [37].

Noteworthy is the laser texturing of fabric surfaces, which increases their functional qualities, and, among others, reduces fabric trends for pilling. Very good results in this field have been obtained by the authors of the article [38]. Texturing can reduce or increase the surface energy of the modified surface. This achieves better results in the resistance and cohesion of the vapor deposited layer. The coalescence phenomena supported by the intended surface texture can also lead to such metallic structures deposited on a modified substrate, whose crystal structure can be better ordered than layers deposited on a similar substrate without such laser modification. The laser enables micro-precise treatment and shaping of necessary elements for forming sensors on fabric and polymer substrates [39,40,41]. The laser beam is also used to modify the surface of composites and polymers [42].

Substrates used for applying thin layers with the PVD process should be pretreated and initially cleaned to ensure better adhesion of the metal to the substrate. Premodification of the substrate is also a technological challenge. There are different pretreatments for substrates, for example, plasma treatment, plasma polymerization, glow discharge plasma, high frequency plasma, fluoridation, corona treatment, etc. We proposed using a laser beam to change the surface energy of a textile composite substrate on which thin metallic layers could be deposited. Such textile substrates can be used as the elements of wearable electronics. The presented method is an alternative to the application of environmentally unfriendly chemicals used during conventional pretreatments such as anodizing or etching.

The aim of laser modification of fabric surfaces is to smooth the surface of the fabric, and therefore obtain the requested adhesion during the metallic layer deposition. The goal of the research was to determine the resistance of the deposited layer with the assumed electrical parameters on the fabric using the modification of the substrate surface. The laser modification of the composite substrate used for the research was intended to change the surface of the polyurethane layer by developing it, thereby changing the geometrical parameters of the deposited layer and resulting in changes of the resistance of the deposited thin electroconductive layer. It was necessary to provide adequate energy during a single pulse to cause local polymer melting. Pulse energy was controlled by controlling the beam power and pulse duration. To avoid the effect of energy accumulation in one place, the beam scans at a specific speed correlated with the pulse repetition frequency. This was also one of the reasons for using pulsed lasers instead of the continuous wave (CW) lasers. The pulse repetition rate was determined by the laser manufacturer for a given pulse duration. As a result of the presented modification, it was possible to create structures that are used as passive elements (resistors, coils), as well as conductive paths and electrodes applicable in textronics and wearable electronics.

The choice of optimal laser parameters (the intensity of the laser beam, number of pulses, and duration of pulses) depends on the modified materials. The failure mode of changes from adhesive to cohesive depends on the used laser pulses parameters. This enables the control of the cohesion energy according to the expected end-use requirements in textronics.

Table 1 presents the parameters of the fiber laser (type SM 20W with the F-theta objective - SPI Lasers Ltd., Southampton, UK) used for the modification of the surface layer before the PVD of silver onto the textile called Cordura.

The process was held in the laboratory stand which is presented schematically in the Figure 4.

Overlapping of successive pulses (*l*) of laser beam radiation results from the velocity of scanning (v) and frequency of the laser pulse (*f*) according to the formula:$$l=\frac{v}{f}$$

Figure 5 shows two kinds of overlapping used in the experiment.

The mutual correlation of the pulse repetition frequency and beam scanning speed affects the mutual overlap of the pulse interaction surface. At a smaller value of beam scanning speed, the edge of the treated surface is more equal as compared with the value of a higher speed. However, in this case the beam interacts repeatedly in a given area. For 80 kHz frequency and 400 mm/s speed, the overlapping factor is 5 µm, and for a ten times higher velocity, the coefficient is 50 µm.

### 2.3. Experimental Planning

Experimental planning is used in situations where many factors can affect the results of experimental research. In the case of processes that are dependent on, for example, five input factors and five degrees of variation, 3125 measurements should be made which is time consuming and sometimes impossible to perform. Experimental planning involves selecting the smallest possible number of combinations of input factors on the basis of which a series of experimental trial tests are performed and a mathematical model is built which allows further modification to optimize the process.

Experimental planning techniques were used to assess the impact of individual laser process parameters on the resistance of the conductive layers. Variable factors that can affect the machining quality include the following: pulse repetition rate, beam power, and scanning speed. The duration of the pulse and hatching did not change (15 ns, 0.01 mm). The following ranges of changes in the input parameters were determined based on the preliminary experimental research: pulse repetition frequency 200 to 290 kHz, scanning speed 400 to 4000 mm/s, and power in two ranges, i.e., 1–4 W and 4–6 W (two separate models). The output factor is the resistance *R* of the conducting layer evaporated on the fabric. For input factors in a given range of changes, units of variation and central values were calculated according to the following relationships:$$∆x_{1}=\frac{x_{1max}-x_{1min}}{2}$$
$$x_{10}=\frac{x_{1max}+x_{1min}}{2}$$
where *x*_1_ is the unit of variation, *x*_1max_ is the maximum value of the first variable, *x*_1min_ is the minimum value of the first variable, and *x*_10_ is the central value of the first variable. Input factors were coded according to the following scheme:$$x_{1}=\frac{f-x_{10}}{∆x_{1}}$$
$$x_{2}=\frac{v-x_{20}}{∆x_{2}}$$
$$x_{3}=\frac{p-x_{30}}{∆x_{3}}$$
where *x*_1_ is the pulse repetition frequency *f* (kHz), *x*_2_ is the beam scanning speed *v* (mm/s), and *x*_3_ is the beam power *p* (W). The output factor *R* (Ω) is coded as *y_i_*. A set of measurements and calculations is presented in Table 2 and Table 3.

The sign “+” in Table 2 and Table 3 indicates the maximum value from the adopted range for a given variable, the sign “–” indicates the minimum value. According to the adopted plan, a set of eight samples was made on which the conductive layers were evaporated, and their resistance was measured (*R*_1_, *R*_2_, and *R*_3_). The average resistance value *R*_av_ and the mean square deviation *S*^2^(*y*)*_i_* were calculated and:$$S^{2}{\left(y\right)}_{i}=\frac{\sum_{i=1}^{r}{\left(y_{ui}-y_{śr}\right)}^{2}}{r-1}$$
where *r* is the number of repetitions of measurements of the output quantity and *y*_av_ is the average value of the output quantity.

For the performed tests, the repeatability of the measurement results was determined using the Cochran test. The Cochran coefficient for the performed measurements was calculated based on the formula:$$G=\frac{S^{2}{\left(y\right)}_{i\;max}}{\sum_{i=1}^{N}S^{2}{\left(y\right)}_{i}}$$
where *N* is the number of tests and *S*^2^(*y*)*_i_*_max_ is the maximum value. The value of the critical coefficient was determined based on the Cochran table for *f*_1_ = *N* = 8, *f*_2_ = *r*−1 = 3−1 = 2, and *G_kr_* = 0.5157. In both cases, the calculated factor is less than the critical values (*G*_1_ = 0.4 and *G*_2_ = 0.46) which means the repeatability of the measurements which were made.

The individual regression coefficients for individual variables were calculated on the basis of Table 2 and Table 3 and the equation:$$b_{0}=\frac{1}{N}\overset{N}{\underset{i=1}{\sum }}x_{0i}y_{i}$$
where *x*_0*i*_ is the sign “+” or “–” according to Table 2 and Table 3 for the corresponding variable and *y_i_* is the average resistance value. The values of individual coefficients are collected in Table 4.

For each coefficient, the significance of the influence of a given input variable (frequency, speed, and power) on the value of the resulting factor (resistance) was checked by comparing the values of the calculated coefficients with the critical value:$$b_{kr}=t_{kr}\sqrt{\frac{S^{2}\left(y\right)}{Nr}}=2.12\sqrt{\frac{0.027}{8\cdot 3}}=0.07$$
where *t_kr_* = 2.12 from the Student’s t-test table for  f=N(r−1)=8(3−1)=16.

The comparison shows that only the b_1_ coefficient for the range 1–4 W is not significant, which means that the frequency of repetition of pulses in the range of power changes from 1 to 4 W does not affect the value of the resistance of the deposited layer. Individual regression equations describing the dependence of significant input factors on the value of the resistance of the evaporated layers after decoding are as follows:$$y=2.17+0.095x_{2}-0.275x_{3}=2.17+0.095\frac{v-2200}{1800}-0.275\frac{p-2.5}{1.5}$$
$$y=3.32-1.38x_{1}+1.12x_{2}+1.48x_{3}=3.32-1.38\frac{f-245}{45}+1.12\frac{v-2200}{1800}+1.48\frac{p-2.5}{1.5}$$

It was concluded from the analysis that the frequency of repetition of pulses had a negligible effect on changes in the resistance of the produced conductive layers, therefore, during detailed measurements, this parameter was completely omitted. The power of laser radiation has the greatest impact on changing the resistance of conductive layers. Detailed experimental research was expanded to include further values of this parameter. The effect of beam scanning speed was analyzed from the overlapping point of view. The tests were carried out for the previously adopted two scanning speeds of 400 and 4000 mm/s, in which there is a strong and a weak effect of overlapping successive pulses.

## 3. Results

Using the results of the conducted research, we assessed the thickness of the deposited layer in the PVD process. The profile of the film is presented in Figure 6.

Estimation of the thickness of the metallic layer produced on a flexible substrate is only possible for indirect measurements. During the technological process, laboratory glass was placed next to the modified and tested substrates. Then, the thickness of the layer was assessed by measuring the step height of the measuring needle between the glass and the deposited layer. The measurement was taken across the created test structure. The Dektak 3ST profilometer (Veeco, New York, NY, USA) was used to determine the thickness of the metallic layer on the test glass. The results were considered equivalent to the thickness of the layer on the composite substrate. The layer thickness was determined at 200 nm (Figure 6).

The quality of the produced layer was assessed on the basis of scanning electron microscope (SEM) tests. An example microscopic photo is presented in Figure 7.

The SEM image was obtained using an Hitachi S-4200 scanning microscope (Tokyo, Japan) with 400x magnification. The photo shows a homogeneous metallic layer with small cracks on the surface that have a small effect on the layer resistance [43,44].

During the experiment, different influences of laser parameters were observed. As a result of the tests conducted on a series of samples, the effect of laser micro treatment was found to be dependent on the energy of the laser beam, the frequency and duration of pulses, and the scanning velocity of the laser beam. The surface texture caused by the interaction of laser radiation strongly depends on the mutual placement of subsequent impulses, i.e., both the distance of subsequent scanning paths and the overlapping in relation to the diameter of the laser beam. This possibility of selective and fully-controlled surface modifications before the vacuum deposition process favors the precision of laser technologies as compared with plasma technologies. The number of parameters influencing the final result of micromachining with laser radiation also indicated the ability to control the adhesion and cohesion energy in such processes, consistent with the consumer’s requirements. These effects were observed for each value of energy from 11 to 88 μJ. Although the changes, in the Cordura used as the substrate for silver deposited layer, were hardly visible without magnification for the small value of energy, the effect of such treatment decreased the resistance 2.5 times in relation to the surface resistance of deposited layer on the unmodified surface. The required effect of such a laser micro treatment should be obtained by such parameters which do not damage the mechanical strength of the substrate due to an absorbed energy value that is too great. This was confirmed by the investigation tests conducted using the energy of laser pulses from 11 μJ to 88 μJ. The maximum used energy caused damage to the surface but the layered structure of the substrate persisted. The formation of the layer during the PVD process depends on the physical and chemical properties of the substrate. The surface energy plays a special role in absorbing metal vapors, their movement on the substrate, and coalescence leading to the formation of the island structure, and then the continuous layer. This surface energy was modified by laser micro treatment because it is strongly dependent on the obtained roughness of its surface. A picture of such roughness is presented in Figure 8.

Surface morphology was characterized based on observations using a confocal microscopy (CM). The surface parameters determined in the CM tests had a direct influence on the surface energy measurements of the material. They also determined the mechanism of formation of the metal layer in the first phase of the process of physical vapor deposition of metal vapors.

As a result of the interaction of the laser beam with the Cordura, the structure of the polymer layer located on the surface of polyamide fibers changed. The choice of laser beam power was associated with the expected final effects on the form of the created layers’ resistance. The laser beam only modified the polymer placed on the surface of the textile composite without affecting the structure of the interwoven threads used as the base part in the tested composite.

The effect of the laser beam influence on the Cordura can be seen in photos taken using a Delta Optical SZ 630-T optical microscope. The photos were taken at a magnification of 60×.

Figure 9a shows the surface of Cordura without any modification. The image presented in Figure 9b shows the surface of Cordura after modification using a 4 W laser beam with 400 mm/s scanning speed. Local melted polymer is visible on the surface. Figure 9c shows the surface of a composite subjected to a laser beam with a power of 6 W. The laser beam exposure time was 15 ns. As a result of the action of a laser beam with such power, the surface layer of Cordura was destroyed. There were numerous cracks in the polymer layer. The resistance of the metallic layer deposited on such a surface is more than five times higher than the lowest value. The laser pulses did not use a large amount of energy. Focusing the laser beam at 26 μm released 10 mJ of energy in a small volume that was capable of melting the surface polymer of Cordura and forming very small craters and defects on its surface which influenced the resistance of the metallic deposited layer. It is possible to change the surface roughness of Cordura and at the same time the resistance of the deposited layers by using the proper parameters of the laser.

As a result of the described laser modification of the polyurethane layer (the surface layer of the applied composite substrate), the increase or decrease in the resistance of deposited metal layer measured by the four-probes method was observed (Figure 10). Conducting the laser modifications increased the surface resistance resulting in reduced surface energy of the surface being treated. During the conducted research, two values of scanning velocity were taken into account, i.e., 400 and 4000 mm/s which resulted in two values of overlapping of successive pulses of laser beam radiation. As previously mentioned, for a lower velocity, the areas of impact of the laser beam overlapped to a greater extent. As a result, greater homogeneity in the area of surface modification is achieved. The action of the laser beam on the polyurethane substrate made the changes in the surface structure. They effect the surface energy of the substrate, and hence the resistances of the deposited layers. In the initial phase of the modification, a decrease in the resistance value is observed. It results both from the removal of impurities from the surface of the substrate, as well as from the change of its surface energy. The resistance of the deposited layers decreases, but after exceeding a certain value of the laser pulse energy, an increase of the resistance of the applied silver layer is observed due to the destruction of the polyurethane layer. After exceeding 4 W at 400 mm/s and 6 W, respectively, at 4000 mm/s, the surface layer of the substrate partially melts, which causes its additional deformation and changes the surface structure of Cordura. Therefore, it is pointless to further increase the power of the laser beam to modify the treated textile composite. It should be noted that with lower energy values of the laser pulse, the resistance increases, and according to the increase of energy the value of resistance decreases until the minimum value is reached. During the process, the distance between adjacent texture unevenness vertices depends on the energy of the absorbed laser radiation. If the inequalities correspond to the total multiple of the solid silver lattice, the resulting structure is less defective and the resistance decreases. In the case of the least favorable, i.e., an odd multiple of half of the grid, the defects are much higher and the resistance increases.

Pulse energy is controlled by setting the beam power and pulse duration. To avoid the effect of energy accumulation in one place, the beam is scanned at a specific speed correlated with the pulse repetition frequency. This ensures better precision (narrower path of the laser beam trace) for such a pulsed laser as compared with a CW laser. This was the reason why all research was done with a pulsed laser. The pulse repetition rate was determined by the laser manufacturer for a given pulse duration.

The conducted research also concerned the assessment of the impact of other laser beam interaction parameters on the surface structure of the substrate, and thus on the resistance of deposited thin metallic layers. The results of the conducted tests are presented in Figure 11, Figure 12, Figure 13 and Figure 14. Figure 11 presents changes in resistance as a function of the frequency of the laser beam interaction. Other invariable parameters of laser processing are the following: 15 ns laser beam exposure time, 400 mm/s scanning speed, 0.01 mm hatching, and 6 W power. With an increase in the frequency of the laser beam interaction on the tested substrate with a composite, a proportional decrease in the resistance value of the created layers was observed. The lowest resistance was obtained for the highest frequency of 290 kHz which was the frequency limit for the laser used for testing. The difference in resistance varied up to 50%.

During the research work, changes in the resistance of the tested electrically conductive layers as a hatching function were also observed. The impact of this parameter is shown in Figure 12. In this case, only hatching was changed, and the other parameters were constant and equal to the 15 ns laser beam exposure time, scanning speed 4000 mm/s, scanning frequency 290 kHz, and 6 W power. The best results were obtained when the distance between the scanning paths was similar to the diameter of the focused laser beam (26 µm), i.e., for 0.01 and 0.02 mm. If the hatching was larger than the beam diameter, the scanned area was not complete and it resulted in greater resistance.

Another parameter that can be changed during laser modification is the scan speed. The effect of this variable on the resistance of the created structures is shown in Figure 13. In the presented example, i.e., with a 220 ns pulse duration, 0.01 mm hatching, 35 kHz laser beam frequency, and 4W power, the best results were obtained for a scanning speed of 400 mm/s. However, it should be emphasized that this was the best speed for 35 kHz. For such a correlation between these two values, we obtained the most favorable overlapping effect. The optimum speed for 290 kHz was 4000 mm/s.

The value of the resistance of the created layers can also be affected by the pulse duration. The lowest resistance value was obtained for the shortest pulse duration of 15 ns and the longest pulse duration of 220 ns (Figure 14). However, the fabric deformed due to thermal damage to the substrate for longer pulses. The presented results were obtained with the remaining parameters equal to hatching 0.01 mm, beam scanning speed 400 mm/s, and laser beam power 6 W.

## 4. Conclusions

The presented method can be used to produce passive elements used in the design and production of wearable electronic systems. Such an increase is strongly dependent on the obtained texture of the surface layer after the laser processing, which in this way can introduce more or fewer defects into the deposited metallic layer. The decrease of resistance depends on the energy used and on the relation between other parameters mentioned in the article (hatching, overlapping, and scanning velocity). The ability to control the surface energy value, and thus the value of surface resistance is of special utilitarian importance, i.e., in wearable textronics for firemen and for smart sensors incorporated for ill people in hospitals. The quality of the nanosecond ablated surface of Cordura as a substrate for metallic layers, in optimal conditions of processes, is sufficient to improve surface resistance according to the requirements.

## Figures and Tables

**Figure 1 sensors-20-01920-f001:**
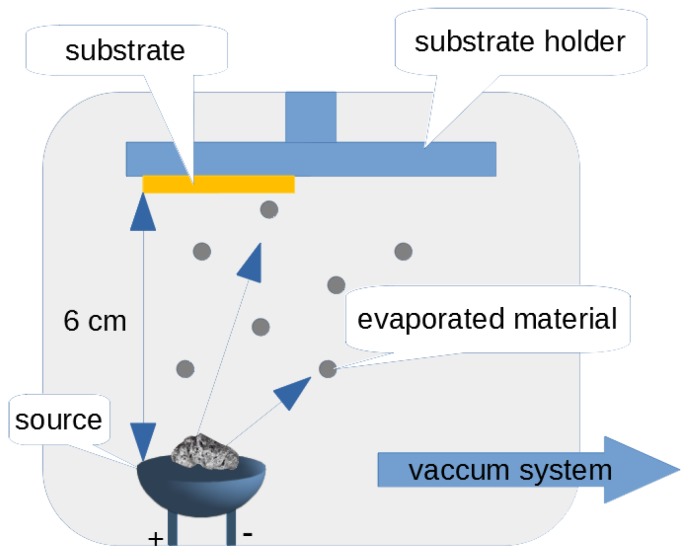
Schematic drawing of the physical vapor deposition (PVD) process, thermal evaporation technique.

**Figure 2 sensors-20-01920-f002:**
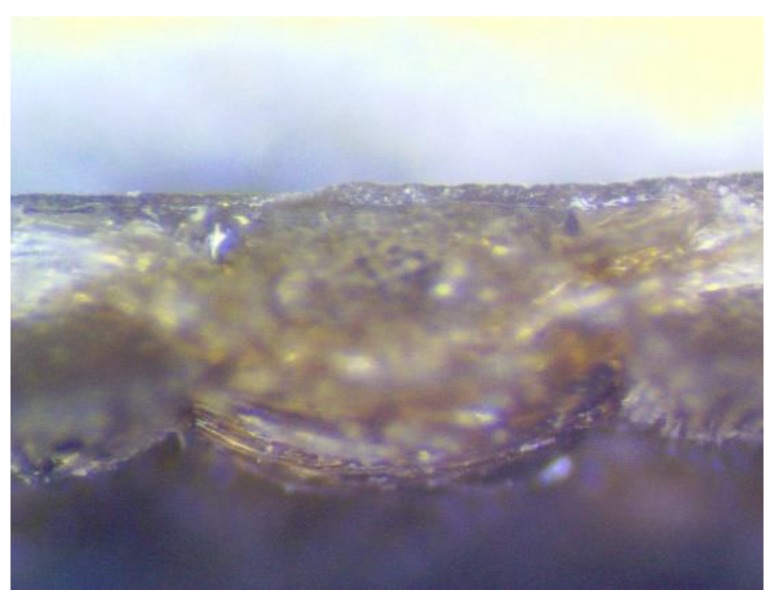
A cross-section of the tested textile composite.

**Figure 3 sensors-20-01920-f003:**
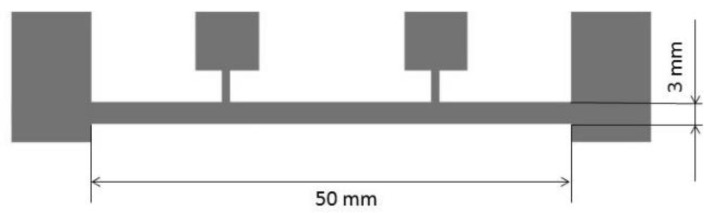
Geometry and dimensions of the test structure created using the PVD process on textile substrates.

**Figure 4 sensors-20-01920-f004:**
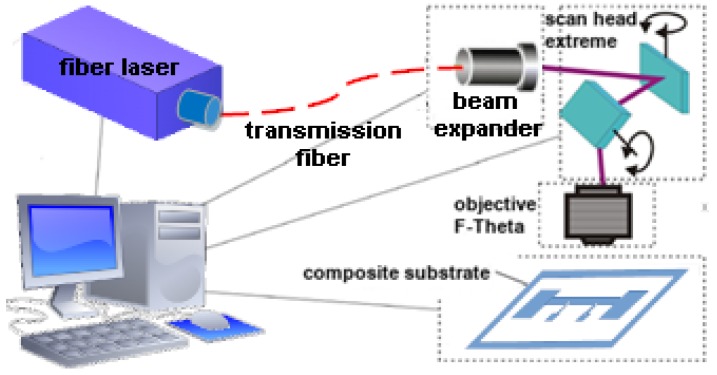
Single mode laser stand with Nutfield scan head and objective F-theta.

**Figure 5 sensors-20-01920-f005:**
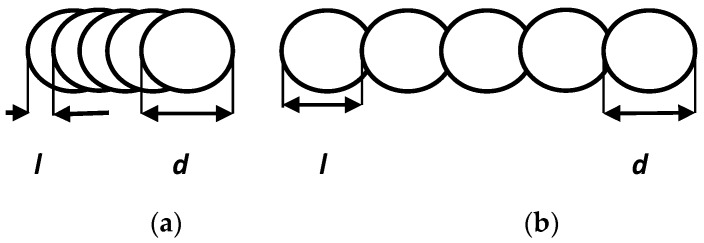
Overlapping of successive pulses generated with a frequency of 80 kHz. (**a**) For ***v*** = 400 mm/s; (**b**) For ***v*** = 4000 mm/s.

**Figure 6 sensors-20-01920-f006:**
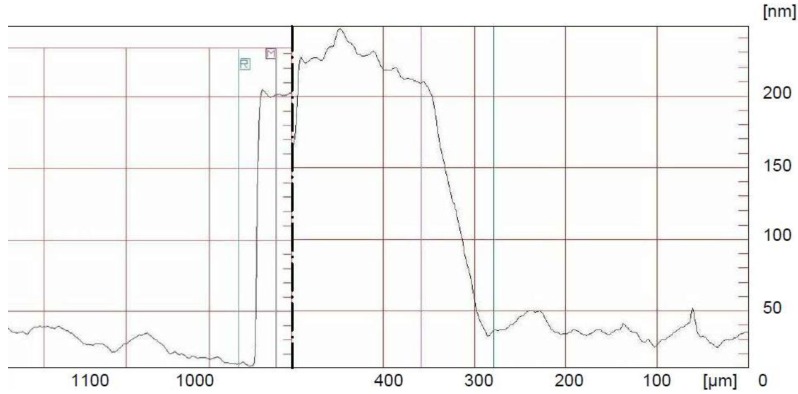
Profile thickness of layers on the edge and in the middle of the cross-section of the test structure.

**Figure 7 sensors-20-01920-f007:**
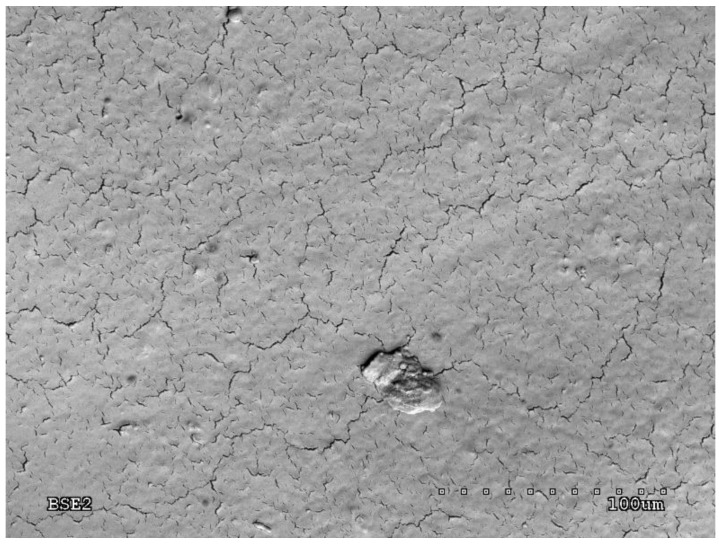
SEM image of the silver layer deposited on the Cordura unmodified substrate.

**Figure 8 sensors-20-01920-f008:**
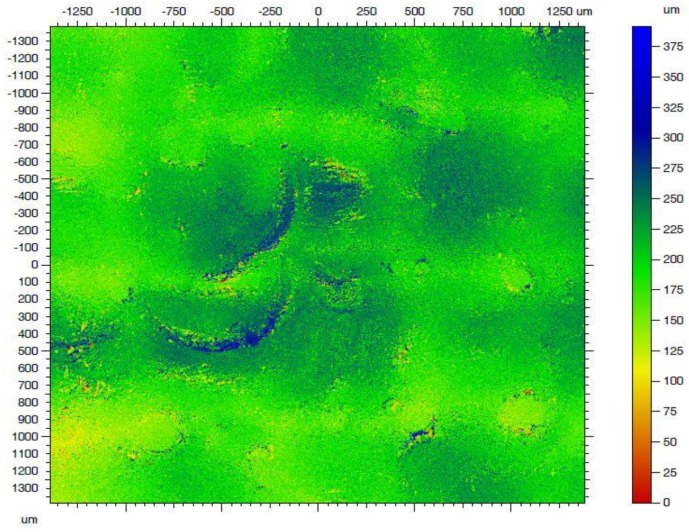
Surface morphology of the Cordura on a micrometer scale (confocal microscopy).

**Figure 9 sensors-20-01920-f009:**
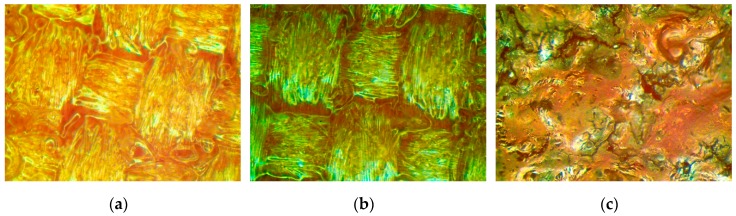
Microscopic images of the surface of the Cordura composite substrate. (**a**) Without modification; (**b**) After laser modification, 4 W beam power; and (**c**) After laser modification, 6 W beam power.

**Figure 10 sensors-20-01920-f010:**
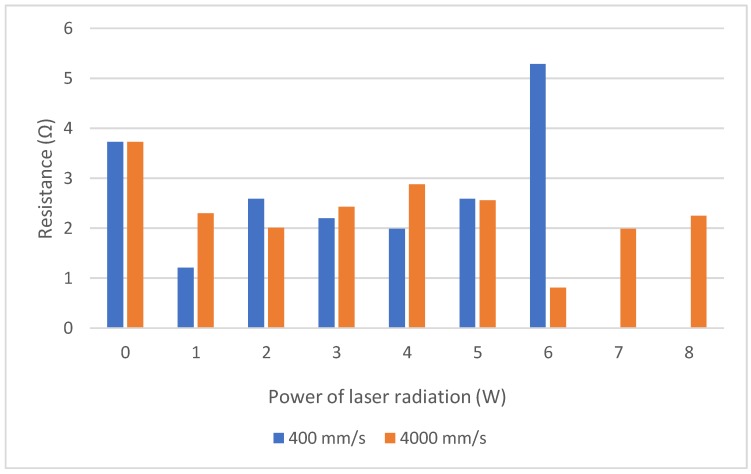
The surface resistance of the thin silver layer as a function of the power of laser radiation.

**Figure 11 sensors-20-01920-f011:**
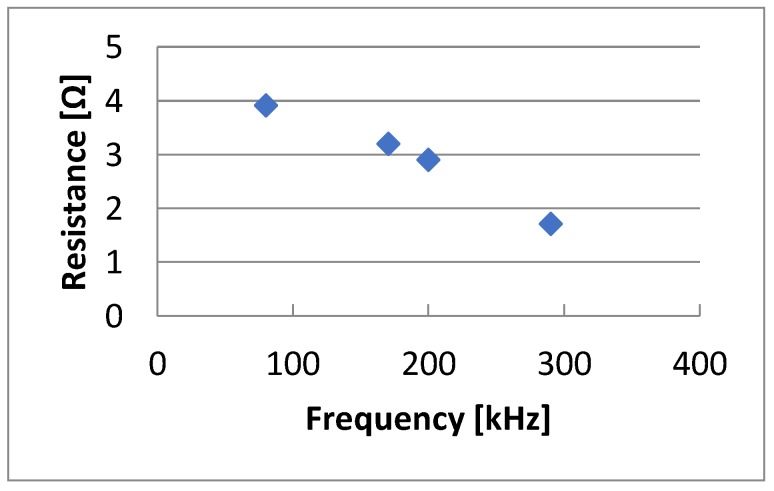
The dependence of resistance changes on the frequency of the laser beam.

**Figure 12 sensors-20-01920-f012:**
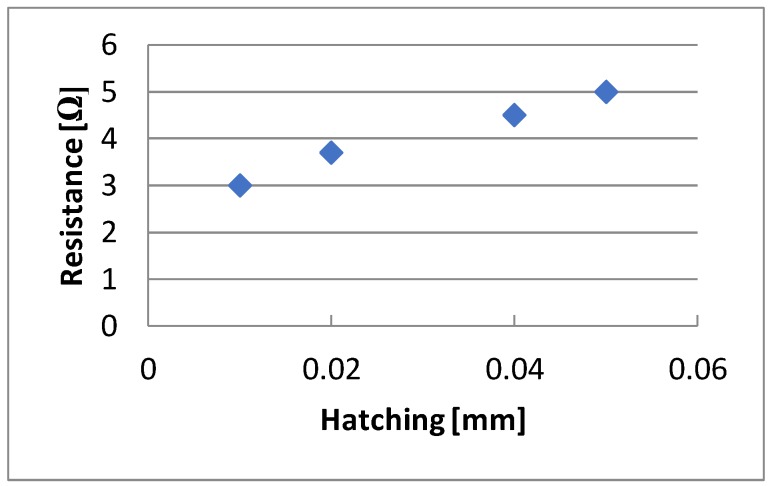
The dependence of resistance changes on hatching.

**Figure 13 sensors-20-01920-f013:**
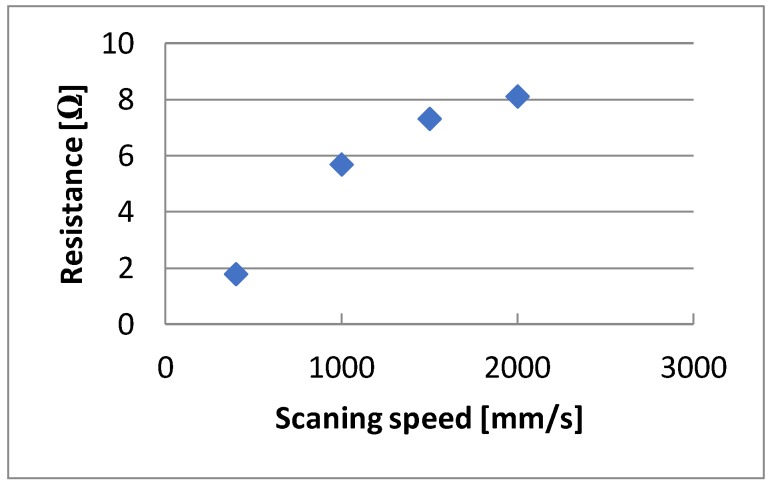
The relationship between resistance changes and scanning speed.

**Figure 14 sensors-20-01920-f014:**
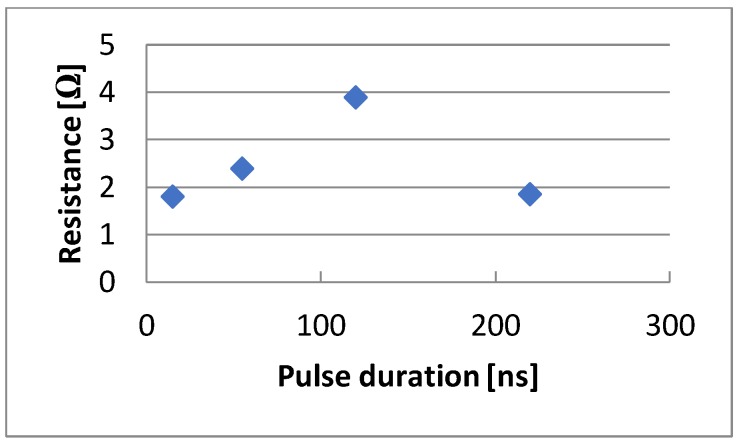
The dependence of resistance changes on the pulse duration.

**Table 1 sensors-20-01920-t001:** Parameters of the fiber laser.

Parameter	Symbol	Unit	Value
pulse energy	*E_p_*	μJ	220 (88)–11
hatching	*h*	mm	0.01–0.04
overlapping	*l*	μm	2 and 20
pulse duration	*τ_p_*	ns	15–220 (55)
diameter of laser beam	*d*	μm	26
frequency of repetition	*f*	kHz	80
scanning velocity	*v*	mm/s	400 and 4000

**Table 2 sensors-20-01920-t002:** Matrix of the experimental plan for power in the range 1–4 W.

No	*x* _0_	*x* _1_	*x* _2_	*x* _3_	*R* _1_	*R* _2_	*R* _3_	*R* _av_	*S*^2^(*y*)*_i_*
1	+	+	+	+	1.8	1.5	1.5	1.6	0.03
2	+	+	+	-	3.3	3.4	3.9	3.53	0.1
3	+	+	-	+	1.5	1.4	1.7	1.57	0.023
4	+	+	-	-	2.1	2	2.7	2.27	0.14
5	+	-	+	+	1.9	1.9	1.9	1.9	0
6	+	-	+	-	1.9	1.9	2.2	2	0.03
7	+	-	-	+	2.6	2.7	2.4	2.57	0.02
8	+	-	-	-	1.9	1.9	1.9	1.9	0

**Table 3 sensors-20-01920-t003:** Matrix of the experimental plan for power in the range 4–6 W.

No	*x* _0_	*x* _1_	*x* _2_	*x* _3_	*R* _1_	*R* _2_	*R* _3_	*R* _av_	*S*^2^(*y*)*_i_*
1	+	+	+	+	2.6	2.7	3.2	2.83	0.1
2	+	+	+	-	1.8	1.5	1.5	1.6	0.03
3	+	+	-	+	1.7	1.8	1.8	1.77	0.003
4	+	+	-	-	1.5	1.4	1.7	1.53	0.02
5	+	-	+	+	11.6	11.4	11.2	11.4	0.04
6	+	-	+	-	1.9	1.9	1.9	1.9	0
7	+	-	-	+	3	2.9	2.9	2.93	0.003
8	+	-	-	-	2.6	2.7	2.4	2.57	0.02

**Table 4 sensors-20-01920-t004:** Summary of regression coefficient values.

Power Value	*b* _0_	*b* _1_	*b* _2_	*b* _3_
1–4 W	2.17	0.075	0.09	−0.257
4–6 W	3.32	−1.38	1.12	1.42

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
