# Peer review of "The Influence of Laser Modification on a Composite Substrate and the Resistance of Thin Layers Created Using the PVD Process"

_sensors, 2020, doi:10.3390/s20071920_

Round 1

Reviewer 1 Report

The authors showed the results of the modification of a textile composite substrate using laser and its influence on the surface resistance of the coating deposited by PVD. It seems interesting that they studied on the effects of different laser processing parameters on the conductivity of the deposited surface layer. However, there is almost no result and discussion about the related coating technique. And the authors were not able to describe even the basic terms of the deposition techniques. For example, PVD stands for the physical vapor deposition and CVD for the chemical vapor deposition Those are very basic terms in physics. Moreover, the manuscript should be completely rewritten or revised. Unfortunately I am not able to accept this manuscript.

Author Response

Dear Reviewer,

We would like to thank you very much for your time you spent for correcting our paper and suggestions. They have helped us to improve the paper.

Below we have written answers for your comments. All of them are placed directly under the comments and they are highlighted in the text :

The authors showed the results of the modification of a textile composite substrate using laser and its influence on the surface resistance of the coating deposited by PVD. It seems interesting that they studied on the effects of different laser processing parameters on the conductivity of the deposited surface layer. However, there is almost no result and discussion about the related coating technique. And the authors were not able to describe even the basic terms of the deposition techniques. For example, PVD stands for the physical vapor deposition and CVD for the chemical vapor deposition Those are very basic terms in physics. Moreover, the manuscript should be completely rewritten or revised. Unfortunately I am not able to accept this manuscript.

The manuscript was improved by the authors according to the reviewers’ suggestions. A lot of additional information according to the reviewers’ suggestions has been added to the text. The information about the PVD process was expanded.

The title has been changed to “Influence of Laser Modification of Composite Substrate on Resistance of Thin Layers Created in PVD Process”

The following text was added into the introduction part:

“Another method of producing thin conductive layers on various substrates is the electrospinning technique [7,8]. As it turns out, this technique is suitable not only for the production of polymer nanofibers but also for the production of conductive layers that are widely used in electronic components and devices [9,10]. Electrospinning is also used to produce electrostatic, conductive nanofibers and transparent conductive layers [11,12].“ was added to the introduction part of the paper.”

The following information has been also added to the manuscript:

“In the article, the authors focused on the production of thin electroconductive layers using physical vacuum deposition as an environmentally friendly process and important in the case of the improvement of corrosion and wear resistance [13]. Using PVD technologies, it is possible to produce coatings with a homogeneous structure as well as layers with graded properties and controlled morphology, coatings with excellent adhesion or with predefined shapes [14–18]. The process of physical deposition of layers can be improved by the results of many works on mathematical modelling and numerical simulations [13, 19, 20]. As a result, it is expected that the costs of carrying out the process will be reduced as well as the impact on the quality and mechanical and electrical properties of the structures produced in this process [21-23]. PVD coatings are also considered as i.e. an alternative replacement of hard chromium plating in the plastic mold industry [24].”

Section 2 has been extended with the following information:

“Evaporation is one of the methods for surface coating in a PVD process. This technique allows the particles of metal which is placed in a heated source to be released and settles them on a substrate located in close proximity [13]. The substrate temperature is much lower than the temperature of boiling metal. A thin layer is formed as a result of atom-on-atom deposition on the prepared substrate. Individual particles of deposited metal are released by thermal evaporation from a resistance source. Thin layers usually have thicknesses of several microns. In the discussed process, the surface properties change and the transition zone between the substrate and the deposited material is observed. The vacuum environment inside the process chamber during deposition is conducive to the elimination of gaseous pollutants, and thus greater chemical homogeneity of the deposited layer. The substrate on which the thin electroconductive layer is deposited also affects the electrical and mechanical properties of the deposited layer [13]. A diagram of the PVD process mechanism is presented in Fig. 1.

Figure 1. Schematic drawing of PVD process – thermal evaporation”

Some other information according to reviewers’ suggestions has also been written in the text:

“The aim of the laser modification of the surface of the fabric is smoothing the surface of the fabric so as to obtain the request adhesion during the metallic layer deposition. The goal of the research is to obtain the resistance of the deposited layer with the assumed electrical parameters on the fabric using the modification of the substrate surface. The laser modification of the composite substrate used for research is intended to change the surface of the polyurethane layer by developing it, thereby changing the geometrical parameters of the deposited layer resulting in changes of the resistance of the deposited thin electroconductive layer. It is necessary to provide adequate energy during a single pulse to cause local polymer melting. Pulse energy is controlled by controlling the beam power and pulse duration. To avoid the effect of energy accumulation in one place, the beam scans at a specific speed correlated with the pulse repetition frequency. This is also one of the reasons for using pulsed lasers instead of the cw lasers. The pulse repetition rate is determined by the laser manufacturer for a given pulse duration. As a result of the presented modification, it is possible to create structures that are used as passive elements (resistors, coils), as well as conductive paths and electrodes applicable in textronics and wearable electronics.”

and the Result part has been also extended with:

“As a result of the interaction of the laser beam with the Cordura, the structure of the polymer layer located on the surface of polyamide fibers changes. The choice of laser beam power is associated with the expected final effects in the form of the created layers resistance. The laser beam only modifies the polymer placed on the surface of the textile composite without affecting the structure of the interwoven threads used as the base part in the tested composite.

The effect of the laser beam inflence on Cordura can be seen in photos taken with the use of Delta Optical SZ 630-T optical microscope. The photos were taken at a magnification of 60x.

Figure 9a shows the surface of Cordura without any modification. Image presented as Figure 9b shows the surface of Cordura after modification by 4 W laser beam with 400 mm/s scanning speed. Local melted polymer is visible on the surface. Figure 9c shows the surface of a composite subjected to a laser beam with a power of 6 W. The laser beam exposure time was 15 ns. As a result of the action of a laser beam with such power, the surface layer of Cordura was destroyed. There are numerous cracks in the polymer layer. The resistance of the metallic layer deposited on such a surface is over five times higher than the lowest value. The used energy of laser pulses is not very large. Focusing the laser beam at 26 μm leads to the release of energy 10 mJ in a small volume capable to melt the surface polymer of Cordura and to form very small craters and defects at its surface. It influences the resistance of the metallic deposited layer. Using the proper parameters of the laser it is possible to change the surface roughness of Cordura and at the same time the resistance of the deposited layers.”

Figure 9. Microscopic images of the surface of the Cordura composite substrate
a) without modification, b) after laser modification - 4 W beam power,
c) after laser modification - 6W beam power

The characteristics which show how other parameters affect the resistance of the metallic thin layer created on the composite textile surface and the description have been added to the manuscript:

“Pulse energy is controlled by setting the beam power and pulse duration. To avoid the effect of energy accumulation in one place, the beam is scanned at a specific speed correlated with the pulse repetition frequency. This ensures better precision (narrower path of the laser beam trace) for such a pulsed laser than for a cw laser. This was the reason why all research was done with pulsed laser. The pulse repetition rate is determined by the laser manufacturer for a given pulse duration.

The conducted research also concerned the assessment of the impact of other laser beam interaction parameters on the surface structure of the substrate and thus on the resistance of deposited thin metallic layers. The results of the conducted tests are presented in Figure 11-14.

Figure 11. The dependence of resistance changes on the frequency of the laser beam

Figure 11 presents changes in resistance as a function of the frequency of the laser beam interaction. Other invariable parameters of laser processing are: 15 ns laser beam exposure time, 400mm / s scanning speed, 0.01mm hatching and 6W power. With the increase in the frequency of the laser beam interaction on the tested substrate with a composite, a proportional decrease in the resistance value of the created layers is observed. The lowest resistance is obtained for the highest frequency 290kHz. This is the frequency limit for the laser used for testing. The difference in resistance varies up to 50%.

Figure 12. The dependence of resistance changes on hatching

During the research work, changes in the resistance of the tested electrically conductive layers as a hatching function were also observed. The impact of this parameter is shown in Figure 12. In this case, only hatching was changed, and the other parameters were constant and equal to: 15 ns laser beam exposure time, scanning speed 4000mm / s, scanning frequency 290 kHz and 6 W power. The best results have been obtained when the distance between the scanning paths is similar to the diameter of the focused laser beam (26µm), i.e. for 0.01 and 0.02mm. If the hatching is larger than the beam diameter, the scanned area is not complete and it results in greater resistance.

Figure 13. The relationship between resistance changes and scanning speed

Another parameter that can be changed during laser modification is the scan speed. The effect of this variable on the resistance of the created structures can be seen in Figure 13. In the presented example, i.e. with a 220 ns pulse duration, 0.01 mm hatching, 35 kHz laser beam frequency and 4W power, the best results are obtained for a scanning speed of 400 mm/s. However, it should be emphasized that this is the best speed for 35kHz. For such a correlation of these two values, we obtain the most favorable overlapping effect. The optimum speed for 290kHz is 4000mm / s.

Figure 14. The dependence of resistance changes on the pulse duration

The value of the resistance of the created layers may also be affected by the pulse duration. The lowest resistance value is obtained for the shortest pulse duration of 15ns and the longest one 220ns (Figure 14). However, the fabric deforms due to thermal damage to the substrate for longer pulses. The presented results were obtained with the remaining parameters equal to: hatching 0.01 mm, beam scanning speed 400 mm/s, laser beam power 6W.”

Some additional information about the thickness and the structure of the deposited layer has been placed in the Results:

“As a result of the conducted research, the authors assess the thickness of the deposited layer in the PVD process. The profile of the film is presented in Figure 6.

Figure 6. Profile thickness of layers on the edge and in the middle
of the cross-section of the test structure:

Estimation of the thickness of the metallic layer produced on a flexible substrate is only possible in indirect measurements. During the technological process, laboratory glass was placed next to the modified and tested substrates. Then the thickness of the layer was assessed by measuring the step height of the measuring needle between the glass and the deposited layer. The measurement was taken across the created test structure. The Dektak 3ST profilometer was used to determine the thickness of the metallic layer on the test glass. The results were considered equivalent to the thickness of the layer on the composite substrate. The layer thickness was determined at 200 nm (Figure 6).

The quality of the produced layer was assessed on the basis of SEM microscopic tests. An example microscopic photo is presented in Figure 7.

Figure 7. Microscopic image of the silver layer deposited on Cordura unmodified substrate

The microscopic image was obtained using the Hitachi S-4200 scanning microscope. 400x magnification was used. The photo shows a homogeneous metallic layer with small cracks on the surface that have a small effect on the layer resistance [41-42].”

Additional references have been also added to the manuscript:

7.Thandavamoorthy Subbiah, G. S. Bhat, R. W. Tock, S. Parameswaran, S. S. Ramkumar, Electrospinning of Nanofibers, Wiley InterScience, 2004.

  1. N. Bhardwaj, S. C. Kundu, Electrospinning: A fascinating fiber fabrication technique, Biotechnology Advances, 28 (2010) 325–347.
  2. J. Miao, M. Miyauchi, T. J. Simmons, J. S. Dordick, R. J. Linhardt, Electrospinning of Nanomaterials and Applications in Electronic Components and Devices, Journal of Nanoscience and Nanotechnology Vol. 10, 5507–5519, 2010.

10.Zexuan Donga, Scott J. Kennedyb, YiquanWua, Electrospinning materials for energy-related applications and devices, Journal of Power Sources 196 (2011) 4886–4904.

  1. Ian D. Norris a, Manal M. Shaker b, Frank K. Ko b, Alan G. MacDiarmid, Electrostatic fabrication of ultrafine conducting fibers: polyanilinerpolyethylene oxide blends, Synthetic Metals 114 2000. 109–114
  2. Hui Wu, Desheng Kong, ZhichaoRuan, Po-Chun Hsu, Shuang Wang, Zongfu Yu, Thomas J. Carney, Liangbing Hu, Shanhui Fan, Yi Cui, A transparent electrode based on a metal nanotrough network, Nature Nanotechnology, 2013.
  3. Baptista A., Silva F., Porteiro J., Míguez J., Pinto G. Sputtering Physical Vapour Deposition (PVD) Coatings: A Critical Review on Process Improvement and Market Trend Demands Coatings 2018, 8, 402; doi:10.3390/coatings8110402
  4. Maity, S. Optimization of processing parameters of in-situ polymerization of pyrrole on woollen textile to improve its thermal conductivity. Prog. Org. Coat. 2017, 107, 48–53.
  5. Kim, M.; Kim, S.; Kim, T.; Lee, D.K.; Seo, B.K.; Lim, C.S. Mechanical and thermal properties of epoxy composites containing zirconium oxide impregnated halloysite nanotubes. Coatings 2017, 7, 231.
  6. Hu, N.; Khan, M.;Wang, Y.; Song, X.; Lin, C.; Chang, C.; Zeng, Y. Effect of Microstructure on the Thermal Conductivity of Plasma Sprayed Y2O3 Stabilized Zirconia (8% YSZ). Coatings 2017, 7, 198.
  7. Silva, F.J.G.; Casais, R.C.B.; Martinho, R.P.A.; Baptista, P.M. Mechanical and tribological characterization of TiB2 thin films. J. Nanosci. Nanotechnol. 2012, 12, 9187–9194.
  8. Silva, F.J.G.; Martinho, R.P.; Andrade, M.; Baptista, A.P.M.; Alexandre, R. Improving the wear resistance of moulds for the injection of glass fibre–reinforced plastics using PVD coatings: A comparative study. Coatings 2017, 7, 28.
  9. Pinto G., Silva F., Porteiro J., Míguez J., Baptista A.: Numerical Simulation Applied to PVD Reactors: An Overview Coatings, 8(11), (2018) 410; https://doi.org/10.3390/coatings8110410
  10. Geiser J, Röhle R: Modelling and Simulation for Physical Vapor Deposition: Multiscale Model Journal of Convergence Information Technology 3, (2008) 4
  11. Voottipruex, P.; Bergado, D.T.; Lam, L.G.; Hino, T. Back-analyses of flow parameters of PVD improved soft Bangkok clay with and without vacuum preloading from settlement data and numerical simulations. Geotext. Geomembr. 2014, 42, 457–467.
  12. Bobzin, K.; Brinkmann, R.; Mussenbrock, T.; Bagcivan, N.; Brugnara, R.; Schäfer, M.; Trieschmann, J. Continuum and kinetic simulations of the neutral gas flow in an industrial physical vapor deposition reactor. Surf. Coat. Technol. 2013, 237, 176–181.
  13. Skordaris, G.; Bouzakis, K.; Kotsanis, T.; Charalampous, P.; Bouzakis, E.; Breidenstein, B.; Bergmann, B.; Denkena, B. Effect of PVD film’s residual stresses on their mechanical properties, brittleness, adhesion and cutting performance of coated tools. CIRP J. Manuf. Sci. Technol. 2017, 18, 145–151
  14. D’Avico L., Beltrami R., Lecis N., Trasatti S. P. Corrosion Behavior and Surface Properties of PVD Coatings for Mold Technology Applications Coatings 2019, 9, 7; doi:10.3390/coatings9010007
  15. Pawłowski S, Plewako J., Korzeniewska E.: Field Modeling the Impact of Cracks on the Electroconductivity of Thin‐Film Textronic Structures, Electronics 9, (2020) 402; doi:10.3390/electronics9030402
  16. Pawłowski S, Plewako J., Korzeniewska E.: Analysis of flow field distribution in a thin conductive layer with an elliptical defect, Przegląd Elektrotechniczny, 96 (2020), 1,:236-239

All numbers of figures and references have been corrected.

Attached please find the improved and upgraded text of our article.

Reviewer 2 Report

In this paper, E. Korzeniewska et al. investigated how the modification of a textile composite substrate affects the surface resistance of silver structures intended for the use in wearable electronics. My comments are stated below:

Point (1) My recommendation is to use another title for the manuscript.

Point (2) The introduction of the manuscript is very short. The authors should specify clearly which are the main innovations and novelties introduced in this manuscript. Apart from that, the authors should report about the current state of the art of related publications of physical vapor deposition technology (PVD).

Point (3) The authors should include a schematic description of the PVD process explaining the intrinsic mechanisms of this fabrication process.

Point (4) Table 1 presents the parameters of the fibre laser used in the modification of the surface layer. Briefly, how these parameters can affect the morphology of the layer. For instance, what are the expected changes of resistance if it is increased the pulse energy or the diameter of the laser beam. 

Point (5) The authors should add an SEM image of the Cordura substrate with the purpose to show the polyamide nanofibers. How is it affecting the morphology of the nanofibers to the resistance of the device? Justify this information in the revised version of the manuscript.

Point (6)  The results given in Figure 6 are not clear. The authors should include the measurement units in the X and Y-axis. 

Point (7) The authors should provide more experimental results. From my perspective, the amount of experimental results given in the paper is not sufficient. For example, it could be investigated how the variation of other processing parameters are affecting the resistance of the substrate. 

Point (8) The techniques of PVD, CVD, and PAVD for deposition of layers are well mentioned in the manuscript. Some references about layers fabricated with the electrospinning technique could be also included to complete the paper.

Author Response

Dear Reviewer,

We would like to thank you very much for your time you spent on correcting our paper and suggestions. They have helped us to improve the paper.

Below we have written answers for your comments. All of them are placed directly under the comments and they are highlighted in the text :

In this paper, E. Korzeniewska et al. investigated how the modification of a textile composite substrate affects the surface resistance of silver structures intended for the use in wearable electronics. My comments are stated below:

Point (1) My recommendation is to use another title for the manuscript.

According to the reviewer’s suggestion, we have changed the title to “Influence of Laser Modification of Composite Substrate on Resistance of Thin Layers Created in PVD Process”

 Point (2) The introduction of the manuscript is very short. The authors should specify clearly which are the main innovations and novelties introduced in this manuscript. Apart from that, the authors should report about the current state of the art of related publications of physical vapor deposition technology (PVD).

The additional text has been added to the Introduction section of the paper:

“In the article, the authors focused on the production of thin electroconductive layers using physical vacuum deposition as an environmentally friendly process and important in the case of the improvement of corrosion and wear resistance [13]. Using PVD technologies, it is possible to produce coatings with a homogeneous structure as well as layers with graded properties and controlled morphology, coatings with excellent adhesion or with predefined shapes [14–18]. The process of physical deposition of layers can be improved by the results of many works on mathematical modelling and numerical simulations [13, 19, 20]. As a result, it is expected that the costs of carrying out the process will be reduced as well as the impact on the quality and mechanical and electrical properties of the structures produced in this process [21-23]. PVD coatings are also considered as i.e. an alternative replacement of hard chromium plating in the plastic mold industry [24]. “

  1. Baptista A., Silva F., Porteiro J., Míguez J., Pinto G. Sputtering Physical Vapour Deposition (PVD) Coatings: A Critical Review on Process Improvement and Market Trend Demands Coatings 2018, 8, 402; doi:10.3390/coatings8110402
  2. Maity, S. Optimization of processing parameters of in-situ polymerization of pyrrole on woollen textile to improve its thermal conductivity. Prog. Org. Coat. 2017, 107, 48–53.
  3. Kim, M.; Kim, S.; Kim, T.; Lee, D.K.; Seo, B.K.; Lim, C.S. Mechanical and thermal properties of epoxy composites containing zirconium oxide impregnated halloysite nanotubes. Coatings 2017, 7, 231.
  4. Hu, N.; Khan, M.;Wang, Y.; Song, X.; Lin, C.; Chang, C.; Zeng, Y. Effect of Microstructure on the Thermal Conductivity of Plasma Sprayed Y2O3 Stabilized Zirconia (8% YSZ). Coatings 2017, 7, 198.
  5. Silva, F.J.G.; Casais, R.C.B.; Martinho, R.P.A.; Baptista, P.M. Mechanical and tribological characterization of TiB2 thin films. J. Nanosci. Nanotechnol. 2012, 12, 9187–9194.
  6. Silva, F.J.G.; Martinho, R.P.; Andrade, M.; Baptista, A.P.M.; Alexandre, R. Improving the wear resistance of moulds for the injection of glass fibre–reinforced plastics using PVD coatings: A comparative study. Coatings 2017, 7, 28.
  7. Pinto G., Silva F., Porteiro J., Míguez J., Baptista A.: Numerical Simulation Applied to PVD Reactors: An Overview Coatings, 8(11), (2018) 410; https://doi.org/10.3390/coatings8110410
  8. Geiser J, Röhle R: Modelling and Simulation for Physical Vapor Deposition: Multiscale Model Journal of Convergence Information Technology 3, (2008) 4
  9. Voottipruex, P.; Bergado, D.T.; Lam, L.G.; Hino, T. Back-analyses of flow parameters of PVD improved soft Bangkok clay with and without vacuum preloading from settlement data and numerical simulations. Geotext. Geomembr. 2014, 42, 457–467.
  10. Bobzin, K.; Brinkmann, R.; Mussenbrock, T.; Bagcivan, N.; Brugnara, R.; Schäfer, M.; Trieschmann, J. Continuum and kinetic simulations of the neutral gas flow in an industrial physical vapor deposition reactor. Surf. Coat. Technol. 2013, 237, 176–181.
  11. Skordaris, G.; Bouzakis, K.; Kotsanis, T.; Charalampous, P.; Bouzakis, E.; Breidenstein, B.; Bergmann, B.; Denkena, B. Effect of PVD film’s residual stresses on their mechanical properties, brittleness, adhesion and cutting performance of coated tools. CIRP J. Manuf. Sci. Technol. 2017, 18, 145–151
  12. D’Avico L., Beltrami R., Lecis N., Trasatti S. P. Corrosion Behavior and Surface Properties of PVD Coatings for Mold Technology Applications Coatings 2019, 9, 7; doi:10.3390/coatings9010007

Point (3) The authors should include a schematic description of the PVD process explaining the intrinsic mechanisms of this fabrication process.

Section 2 has been extended with the following information:

“Evaporation is one of the methods for surface coating in a PVD process. This technique allows the particles of metal which is placed in a heated source to be released and settles them on a substrate located in close proximity [13]. The substrate temperature is much lower than the temperature of boiling metal. A thin layer is formed as a result of atom-on-atom deposition on the prepared substrate. Individual particles of deposited metal are released by thermal evaporation from a resistance source. Thin layers usually have thicknesses of several microns. In the discussed process, the surface properties change and the transition zone between the substrate and the deposited material is observed. The vacuum environment inside the process chamber during deposition is conducive to the elimination of gaseous pollutants, and thus greater chemical homogeneity of the deposited layer. The substrate on which the thin electroconductive layer is deposited also affects the electrical and mechanical properties of the deposited layer [13]. A diagram of the PVD process mechanism is presented in Fig. 1.

Figure 1. Schematic drawing of PVD process – thermal evaporation”

Point (4) Table 1 presents the parameters of the fibre laser used in the modification of the surface layer. Briefly, how these parameters can affect the morphology of the layer. For instance, what are the expected changes of resistance if it is increased the pulse energy or the diameter of the laser beam.

The additional information has been placed into the text:

“The aim of the laser modification of the surface of the fabric is smoothing the surface of the fabric so as to obtain the request adhesion during the metallic layer deposition. The goal of the research is to obtain the resistance of the deposited layer with the assumed electrical parameters on the fabric using the modification of the substrate surface. The laser modification of the composite substrate used for research is intended to change the surface of the polyurethane layer by developing it, thereby changing the geometrical parameters of the deposited layer resulting in changes of the resistance of the deposited thin electroconductive layer. It is necessary to provide adequate energy during a single pulse to cause local polymer melting. Pulse energy is controlled by controlling the beam power and pulse duration. To avoid the effect of energy accumulation in one place, the beam scans at a specific speed correlated with the pulse repetition frequency. This is also one of the reasons for using pulsed lasers instead of the cw lasers. The pulse repetition rate is determined by the laser manufacturer for a given pulse duration. As a result of the presented modification, it is possible to create structures that are used as passive elements (resistors, coils), as well as conductive paths and electrodes applicable in textronics and wearable electronics.”

Point (5) The authors should add an SEM image of the Cordura substrate with the purpose to show the polyamide nanofibers. How is it affecting the morphology of the nanofibers to the resistance of the device? Justify this information in the revised version of the manuscript.

According to the reviewer’s suggestions the microscopic images of the modified substrates with the description have been added to the manuscript:

“As a result of the interaction of the laser beam with the Cordura, the structure of the polymer layer located on the surface of polyamide fibers changes. The choice of laser beam power is associated with the expected final effects in the form of the created layers resistance. The laser beam only modifies the polymer placed on the surface of the textile composite without affecting the structure of the interwoven threads used as the base part in the tested composite.

The effect of the laser beam influence on Cordura can be seen in photos taken with the use of Delta Optical SZ 630-T optical microscope. The photos were taken at a magnification of 60x.

Figure 9. Microscopic images of the surface of the Cordura composite substrate
a) without modification, b) after laser modification - 4 W beam power,
c) after laser modification - 6W beam power

Figure 9a shows the surface of Cordura without any modification. The image presented as Figure 9b shows the surface of Cordura after modification by 4 W laser beam with 400 mm/s scanning speed. Local melted polymer is visible on the surface. Figure 9c shows the surface of a composite subjected to a laser beam with a power of 6 W. The laser beam exposure time was 15 ns. As a result of the action of a laser beam with such power, the surface layer of Cordura was destroyed. There are numerous cracks in the polymer layer. The resistance of the metallic layer deposited on such a surface is over five times higher than the lowest value. The used energy of laser pulses is not very large. Focusing the laser beam at 26 μm leads to the release of energy 10 mJ in a small volume capable to melt the surface polymer of Cordura and to form very small craters and defects at its surface. It influences the resistance of the metallic deposited layer. Using the proper parameters of the laser it is possible to change the surface roughness of Cordura and at the same time the resistance of the deposited layers.”

Point (6)  The results given in Figure 6 are not clear. The authors should include the measurement units in the X and Y-axis.

We absolutely agree with the reviewer’s suggestion.

Figure 6 was replaced with the Figure presented below:

Point (7) The authors should provide more experimental results. From my perspective, the amount of experimental results given in the paper is not sufficient. For example, it could be investigated how the variation of other processing parameters are affecting the resistance of the substrate.

The characteristics which show how other parameters affect the resistance of the metallic thin layer created on the composite textile surface and the description have been added to the manuscript:

“Pulse energy is controlled by setting the beam power and pulse duration. To avoid the effect of energy accumulation in one place, the beam is scanned at a specific speed correlated with the pulse repetition frequency. This ensures better precision (narrower path of the laser beam trace) for such a pulsed laser than for a cw laser. This was the reason why all research was done with a pulsed laser. The pulse repetition rate is determined by the laser manufacturer for a given pulse duration.

The conducted research also concerned the assessment of the impact of other laser beam interaction parameters on the surface structure of the substrate and thus on the resistance of deposited thin metallic layers. The results of the conducted tests are presented in Figure 11-14.

Figure 11. The dependence of resistance changes on the frequency of the laser beam

Figure 11 presents changes in resistance as a function of the frequency of the laser beam interaction. Other invariable parameters of laser processing are 15 ns laser beam exposure time, 400mm / s scanning speed, 0.01mm hatching and 6W power. With the increase in the frequency of the laser beam interaction on the tested substrate with a composite, a proportional decrease in the resistance value of the created layers is observed. The lowest resistance is obtained for the highest frequency 290kHz. This is the frequency limit for the laser used for testing. The difference in resistance varies up to 50%.

Figure 12. The dependence of resistance changes on hatching

During the research work, changes in the resistance of the tested electrically conductive layers as a hatching function were also observed. The impact of this parameter is shown in Figure 12. In this case, only hatching was changed, and the other parameters were constant and equal to: 15 ns laser beam exposure time, scanning speed 4000mm / s, scanning frequency 290 kHz and 6 W power. The best results have been obtained when the distance between the scanning paths is similar to the diameter of the focused laser beam (26µm), i.e. for 0.01 and 0.02mm. If the hatching is larger than the beam diameter, the scanned area is not complete and it results in greater resistance.

Figure 13. The relationship between resistance changes and scanning speed

Another parameter that can be changed during laser modification is the scan speed. The effect of this variable on the resistance of the created structures can be seen in Figure 13. In the presented example, i.e. with a 220 ns pulse duration, 0.01 mm hatching, 35 kHz laser beam frequency and 4W power, the best results are obtained for a scanning speed of 400 mm/s. However, it should be emphasized that this is the best speed for 35kHz. For such a correlation between these two values, we obtain the most favorable overlapping effect. The optimum speed for 290kHz is 4000mm / s.

Figure 14. The dependence of resistance changes on the pulse duration

The value of the resistance of the created layers may also be affected by the pulse duration. The lowest resistance value is obtained for the shortest pulse duration of 15ns and the longest one 220ns (Figure 14). However, the fabric deforms due to thermal damage to the substrate for longer pulses. The presented results were obtained with the remaining parameters equal to: hatching 0.01 mm, beam scanning speed 400 mm/s, laser beam power 6W.”

Point (8) The techniques of PVD, CVD, and PAVD for deposition of layers are well mentioned in the manuscript. Some references about layers fabricated with the electrospinning technique could be also included to complete the paper.

The following text was added into the introduction part:

“Another method of producing thin conductive layers on various substrates is the electrospinning technique [7,8]. As it turns out, this technique is suitable not only for the production of polymer nanofibers but also for the production of conductive layers that are widely used in electronic components and devices [9,10]. Electrospinning is also used to produce electrostatic, conductive nanofibers and transparent conductive layers [11,12].“ was added to the introduction part of the paper.”

Additional references have been also added to the manuscript.

7.Thandavamoorthy Subbiah, G. S. Bhat, R. W. Tock, S. Parameswaran, S. S. Ramkumar, Electrospinning of Nanofibers, Wiley InterScience, 2004.

  1. N. Bhardwaj, S. C. Kundu, Electrospinning: A fascinating fiber fabrication technique, Biotechnology Advances, 28 (2010) 325–347.
  2. J. Miao, M. Miyauchi, T. J. Simmons, J. S. Dordick, R. J. Linhardt, Electrospinning of Nanomaterials and Applications in Electronic Components and Devices, Journal of Nanoscience and Nanotechnology Vol. 10, 5507–5519, 2010.

10.Zexuan Donga, Scott J. Kennedyb, YiquanWua, Electrospinning materials for energy-related applications and devices, Journal of Power Sources 196 (2011) 4886–4904.

  1. Ian D. Norris a, Manal M. Shaker b, Frank K. Ko b, Alan G. MacDiarmid, Electrostatic fabrication of ultrafine conducting fibers: polyanilinerpolyethylene oxide blends, Synthetic Metals 114 2000. 109–114
  2. Hui Wu, Desheng Kong, ZhichaoRuan, Po-Chun Hsu, Shuang Wang, Zongfu Yu, Thomas J. Carney, Liangbing Hu, Shanhui Fan, Yi Cui, A transparent electrode based on a metal nanotrough network,

All numbers of figures and references have been corrected.

Attached please find the improved and upgraded text of our article.

Reviewer 3 Report

Referee Review

The manuscript of Ewa Korzeniewska, Mariusz Tomczyk, Maria Walczak "Changes in the Resistance of Thin-Film Structures Caused by Laser Modification of the Substrate" , Sensors-740765.

The manuscript is devoted to the study of modification of a textile composite substrate (textile Cordura) with the use of laser pulsed irradiation and its influence on the surface resistance of silver layers obtained by the physical vacuum deposition (PVD). The authors suggest that the metallization of textile will contribute to the development of wearable electronics.

The authors use a pulsed laser (1050 nm) with a repetition rate of 80 kHz, an energy of 10-100 mJ, and a laser beam diameter of 26 m. The speed of laser beam along the sample is 400 or 4000 mm/s. A sample was a segment of Cordura 50 mm long and 3 mm wide. After laser treatment a layer of metallic silver was sprayed on the sample in vacuum. Since many factors influence on the electrical resistance of a deposited layer of silver metal, the authors used multivariate analysis to process the obtained results.

The obtained data on electrical resistance have a fairly large scatter, which in all probability is associated with the heterogeneity of Cordura surface. For this reason, the sprayed layer of silver can have gaps, islands and local places without metal. From the value of specific resistivity of silver (~0.015 W mm2/m), it is easy to calculate that the layer of this metal on a flat surface 50×3 mm in size has a resistance of about 2 W (typical resistance obtained in experiments) with a thickness of about 100 nm. Unfortunately, the authors did not determine the thicknesses of the layers of deposited metallic silver. They also did not determine the structure of the sprayed layer of metallic silver.

It seems that for comparison, the authors of manuscript would need to conduct experiments on the deposition of metallic silver on a flat substrate (quartz or silicon plates) with dimensions of 50×3 mm and a deposition time of 5 minutes (as in the experiments with Cordura). In this case, it is possible to determine the resistance and thickness of the flat layer without defects.

Although the energy of laser pulses is not very large, focusing the laser beam at 26 μm leads to the release of energy 10 mJ in a small volume (~10-8 cm3) capable to melt the plastic material of Cordura and to formation of craters (~1000 J/cm3) which then create defects in the deposited layer of metallic silver. It is possible that the laser processing increases the surface roughness of Cordura. The authors did not give a view of the Cordura texture before and after laser treatment. Maybe the authors needed to use a continuous laser to process Cordura and get a smoother surface to deposit the defect-free silver layer.

In line 118, 5 and 50 mm need to replace with 5 and 50 μm

The manuscript could be published in the journal Sensors, however, after taking into account the comments made in the review.

Author Response

Dear Reviewer,

We would like to thank you very much for your time you spent on correcting our paper and suggestions. They have helped us to improve the paper.

Below we have written answers for your comments. All of them are placed directly under the comments and they are highlighted in the text :

The manuscript of EwaKorzeniewska, Mariusz Tomczyk, Maria Walczak "Changes in the Resistance of Thin-Film Structures Caused by Laser Modification of the Substrate" , Sensors-740765.

The manuscript is devoted to the study of modification of a textile composite substrate (textile Cordura) with the use of laser pulsed irradiation and its influence on the surface resistance of silver layers obtained by the physical vacuum deposition (PVD). The authors suggest that the metallization of textile will contribute to the development of wearable electronics.

The authors use a pulsed laser (1050 nm) with a repetition rate of 80 kHz, an energy of 10-100 mJ, and a laser beam diameter of 26 m. The speed of laser beam along the sample is 400 or 4000 mm/s. A sample was a segment of Cordura 50 mm long and 3 mm wide. After laser treatment a layer of metallic silver was sprayed on the sample in vacuum. Since many factors influence on the electrical resistance of a deposited layer of silver metal, the authors used multivariate analysis to process the obtained results.

The obtained data on electrical resistance have a fairly large scatter, which in all probability is associated with the heterogeneity of Cordura surface. For this reason, the sprayed layer of silver can have gaps, islands and local places without metal. From the value of specific resistivity of silver (~0.015 W mm2/m), it is easy to calculate that the layer of this metal on a flat surface 50×3 mm in size has a resistance of about 2 W (typical resistance obtained in experiments) with a thickness of about 100 nm. Unfortunately, the authors did not determine the thicknesses of the layers of deposited metallic silver. They also did not determine the structure of the sprayed layer of metallic silver.

It seems that for comparison, the authors of manuscript would need to conduct experiments on the deposition of metallic silver on a flat substrate (quartz or silicon plates) with dimensions of 50×3 mm and a deposition time of 5 minutes (as in the experiments with Cordura). In this case, it is possible to determine the resistance and thickness of the flat layer without defects.

According to the reviewer’s suggestions the following text and figures have been added to the manuscript:

“As a result of the conducted research, the authors assess the thickness of the deposited layer in the PVD process. The profile of the film is presented in Figure 6.

Figure 6. Profile thickness of layers on the edge and in the middle
of the cross-section of the test structure:

Estimation of the thickness of the metallic layer produced on a flexible substrate is only possible in indirect measurements. During the technological process, laboratory glass was placed next to the modified and tested substrates. Then the thickness of the layer was assessed by measuring the step height of the measuring needle between the glass and the deposited layer. The measurement was taken across the created test structure. The Dektak 3ST profilometer was used to determine the thickness of the metallic layer on the test glass. The results were considered equivalent to the thickness of the layer on the composite substrate. The layer thickness was determined at 200 nm (Figure 6).

The quality of the produced layer was assessed on the basis of SEM microscopic tests. An example microscopic photo is presented in Figure 7.

Figure 7. Microscopic image of the silver layer deposited on Cordura unmodified substrate

The microscopic image was obtained using the Hitachi S-4200 scanning microscope. 400x magnification was used. The photo shows a homogeneous metallic layer with small cracks on the surface that have a small effect on the layer resistance [41-42].”

Additional references are added to the paper:

  1. Pawłowski S, Plewako J., Korzeniewska E.: Field Modeling the Impact of Cracks on the Electroconductivity of Thin‐Film Textronic Structures, Electronics 9, (2020) 402; doi:10.3390/electronics9030402
  2. Pawłowski S, Plewako J., Korzeniewska E.: Analysis of flow field distribution in a thin conductive layer with an elliptical defect, Przegląd Elektrotechniczny, 96 (2020), 1,:236-239

Although the energy of laser pulses is not very large, focusing the laser beam at 26 μm leads to the release of energy 10 mJ in a small volume (~10-8 cm3) capable to melt the plastic material of Cordura and to formation of craters (~1000 J/cm3) which then create defects in the deposited layer of metallic silver. It is possible that the laser processing increases the surface roughness of Cordura. The authors did not give a view of the Cordura texture before and after laser treatment. Maybe the authors needed to use a continuous laser to process Cordura and get a smoother surface to deposit the defect-free silver layer.

The following information has been added to the manuscript:

“As a result of the interaction of the laser beam with the Cordura, the structure of the polymer layer located on the surface of polyamide fibers changes. The choice of laser beam power is associated with the expected final effects in the form of the created layers resistance. The laser beam only modifies the polymer placed on the surface of the textile composite without affecting the structure of the interwoven threads used as the base part in the tested composite.

The effect of the laser beam influence on Cordura can be seen in photos taken with the use of Delta Optical SZ 630-T optical microscope. The photos were taken at a magnification of 60x.

Figure 9. Microscopic images of the surface of the Cordura composite substrate
a) without modification, b) after laser modification - 4 W beam power,
c) after laser modification - 6W beam power

Figure 9a shows the surface of Cordura without any modification. Image presented as Figure 9b shows the surface of Cordura after modification by 4 W laser beam with 400 mm/s scanning speed. Local melted polymer is visible on the surface. Figure 9c shows the surface of a composite subjected to a laser beam with a power of 6 W. The laser beam exposure time was 15 ns. As a result of the action of a laser beam with such power, the surface layer of Cordura was destroyed. There are numerous cracks in the polymer layer. The resistance of the metallic layer deposited on such a surface is over five times higher than the lowest value. The used energy of laser pulses is not very large. Focusing the laser beam at 26 μm leads to the release of energy 10 mJ in a small volume capable to melt the surface polymer of Cordura and to form very small craters and defects at its surface. It influences the resistance of the metallic deposited layer. Using the proper parameters of the laser it is possible to change the surface roughness of Cordura and at the same time the resistance of the deposited layers.”

and

“Pulse energy is controlled by setting the beam power and pulse duration. To avoid the effect of energy accumulation in one place, the beam is scanned at a specific speed correlated with the pulse repetition frequency. This ensures better precision ( narrower path of the laser beam trace) for such a pulsed laser than for a cw laser. This was the reason why all research was done with a pulsed laser. The pulse repetition rate is determined by the laser manufacturer for a given pulse duration.”

In line 118, 5 and 50 mm need to replace with 5 and 50 μm

It has been corrected in the manuscript.

The manuscript could be published in the journal Sensors, however, after taking into account the comments made in the review.

All numbers of figures and references have been corrected.

Attached please find the improved and upgraded text of our article.

Round 2

Reviewer 1 Report

Dear authors,

This manuscript is a lot improved after the authors revised it. This paper could be published after the authors correct at least the following basic scientific terms I mentioned previously.

  • "physical vacuum deposition" should be replaced by "physical vapor deposition"
  • "chemical vacuum deposition" should be replaced by "chemical vapor deposition"

I am not able to understand where those terms come from. Please correct them over the entire manuscript.

In addition, there are more suggestions to be improved.

In the line 23-24, "The challenges faced by modern clothing are not only to provide thermal comfort or to use it as a decorative element. The use of textiles depends on their functionalization." should be rewritten. It is not understandable.

In the line 82, "In the case of the conducted studies" , What does "the conducted studies" mean here? This is not common expression in English. Please rewrite this sentence.

I really recommend the authors let the manuscript check by native speaker.

Author Response

Dear Reviewer,

We would like to thank you very much for the time you spent on correcting our paper and suggestions. The article was corrected as recommended.

Reviewer 2 Report

The manuscript has been significantly improved. I recommend this paper for publication in the journal of sensors. 

Author Response

Dear Reviewer,

We would like to thank you very much for the time you spent on correcting our paper and suggestions

Reviewer 3 Report

Referee Review

The revised manuscript of Ewa Korzeniewska, Mariusz Tomczyk, Maria Walczak "Changes in the Resistance of Thin-Film Structures Caused by Laser Modification of the Substrate" , Sensors-740765.

The authors have significantly improved the manuscript. They took into account all my comments and added new Figures to the manuscript. Most importantly, they measured the thickness of the silver layer. I think that this modified version of the manuscript can be published in the journal Sensors.

Author Response

(The authors gave the same response as above.)
